# Robust Federated Learning Frameworks Guarding Against Data Flipping Threats for Autonomous Vehicles

## Abstract

Federated Learning (FL) has become an established technique to facilitate privacy-preserving collaborative training across a multitude of clients. The ability to achieve collaborative learning from multiple parties containing an extensive volume of data while providing the essence of data privacy made it an attractive solution to address numerous challenges in sensitive data-driven fields such as autonomous vehicles (AVs). However, its decentralized nature exposes it to security threats, such as evasion and data poisoning attacks, where malicious participants can compromise training data. This paper addresses the challenge of defending federated learning systems against data poisoning attacks specifically focusing on data-flipping techniques in AVs by proposing a novel defense mechanism that combines anomaly detection with robust aggregation techniques. Our approach employs statistical outlier detection and model-based consistency checks to filter out compromised updates before they affect the global model. Experiments on benchmark datasets show that our method significantly enhances robustness by preventing nearly 15% of accuracy drop for our global model when confronted with a malicious participant and reduction the the attack success rate even when dealing with 20% of poisoning level. These findings provide a comprehensive solution to strengthen FL systems against adversarial threats.

## 1 Introduction

Autonomous driving Rajasekhar & Jaswal (2015)Martínez-Díaz & Soriguera (2018) is a rapidly developing field that has the potential to revolutionize human transportation. the usage of machine learning in this field proved to by vary promising for it's application in various application, such as autonomous driving Zhang et al. (2016), complex environment navigation Nguyen et al. (2020) , lane following and switching Gurghian et al. (2016) and traffic calculation Yu et al. (2021). Recent advances in this area rely heavily on machine learning, that requires extensive training data. the centralized training provide more accuracy for autonomous driving solutions by using already known and controlled data. This approach neglect data privacy and third party involvement protocols. to confront those issues Federated learning provide multiple advantages.

Federated Learning (FL) Kairouz et al. (2021) McMahan et al. (2017) Chaabene et al. (2022) has become increasingly popular in machine learning, enabling models to learn from distributed devices in diverse contexts. In the FL framework, participating peers undertake the task of training a global model that they receive from the central server using their own local datasets. After processing their local data, these peers generate model updates, which they then send back to the server. The server's role is to gather and aggregate the various updates it receives from all the peers, ultimately leading to the creation of an enhanced global model. Following this aggregation process, the updated global model is redistributed to the peers, setting the stage for the next iteration of training. One of the significant advantages of federated learning lies in its ability to enhance privacy. By keeping the local data on the devices of the peers and not transferring it to a central server, FL significantly mitigates the risks associated with data breaches and privacy violations. This is particularly crucial in an era where data privacy concerns are paramount. Additionally, FL contributes to scalability by distributing the computational workload across the devices of the peers, such as smartphones and other mobile devices, rather than relying solely on a central server's computing resources. This

decentralized approach not only improves efficiency but also allows for a broader range of devices to participate in the training process, making FL a highly attractive option for developing machine learning models in a secure and scalable manner Bonawitz et al. (2019).

But such as any technique FL faces many challenges and limitations; it's distributed nature exposes the global models to potential attacks from malicious participants. The lack of control of the service side Li et al. (2020), encourage malicious behaviour of the peers to tamper with the training guidelines and conduct adversarial attack. Data poisoning represent one of the biggest challenges in the usage of FL, the attackers take advantages of the model distribution to inject misclassified data aiming to make the model fail or not converge, they want to make the model incorrectly classify test cases with particular traits into some desired labels. One sort of targeted poisoning attack is the label-flipping (LF) attack Biggio et al. (2012), in which the attackers tamper with training data in the local model by flipping the labels of select accurate instances from a transfer information from a source class to a target class. Attackers train their local models after contaminating it using the same hyper-parameters, loss function that that the server has supplied for the model architecture . Thus, altering the training data is all that is needed to carry out the assault. That poisoned model is later on send to the server to aggregate with the other intact model, causing a drop of the overall performance and accuracy.

Multiple studies has been conducted to address those issues. Li et al. (2021) investigate peers that have the same goals as attackers, which causes a large percentage of false positives when sincere peers have comparable local data. Zhou et al. (2023) present RoHFL, a hierarchical federated learning framework for the Internet of Vehicles that uses similarity-based reputation scoring and logarithm-based normalization to thwart poisoning assaults. Nevertheless, combining these techniques entangle the aggregate procedure. The OQFL framework Yamany et al. (2023) uses quantum-behaved particle swarm optimization (QPSO) to modify hyperparameters in order to identify hostile cars. However, because the model must be reinitialized and retrained from start, there is a large computational overhead every time a search is conducted.

Our experiments, conducted on image classification datasets such as A2D2 Dataset, CIFAR-10 and a custom image collection used while training our AVs. We use Principal Component Analysis (PCA) to reduce the dimensionality of update vectors and effectively differentiate between malicious and legitimate updates. And we combined it with Multiclass Classification Using Support Vector Machines (SVM). Using PCA, the model can identify anomalies in the distribution of principal components that may indicate inconsistencies in labeling. Once the data is transformed, SVM can be used to classify samples based on the extracted features. SVM decision thresholds can reveal patterns that indicate that some labels are inconsistent with the feature distribution. This may identify flipped labels.,Furthermore, the ensemble method increases the robustness,of the search process. This is because merging multiple SVM classifiers can reveal inconsistencies arising from additional labeling. It helps to be confident. that the model is resilient to adversarial attacks while maintaining high classification performance... Our evaluations on the auxiliary datasets, demonstrate that our defense strategy can effectively identify and block malicious participants.

## 2 RELATED WORK

**Autonomous driving system based on federated learning.** As participatory driving models continue to improve in statistical accuracy, increased attention has been paid to improving the safety and effectiveness of their training programs. Traditionally, driving intervention models have relied on centralized training methods. However, centralized training presents challenges such as server computing capacity limitations, data security concerns, and network transmission overhead Yaacoub et al. (2023). In response, the FD Framework has developed the capacity to nurture these models. FL in Autonomous driving systems has been the subject of a verity of research investigations for various purposes Chellapandi et al. (2023). FL is used, for example, in object detection, it makes it possible for the AV framework to learn quickly and with little communication overhead, which is especially useful when the amount of data is significantly more than the size of the ML model while also protecting the data's privacy. In Barbieri et al. (2022), LiDAR on CAVs is utilized for object classification through a decentralized FL approach. Through V2V networks, the ML model's parameters are exchanged. Comparing FL to selflearning techniques, it has been experimentally demonstrated that FL is highly effective. The identification and recognition of license plates is a

significant additional use of FL. Applications include traffic safety and infractions, traffic monitoring, detecting unlawful or over-time parking, and parking access authentication are only a few of the uses for it in ITS. It has been demonstrated that ML approaches are quite effective in identifying license plates and detecting objects Kong et al. (2021)Xie et al. (2023). The Transformer model has demonstrated the efficacy of the FL framework in learning spatio-temporal characteristics Zhou et al. (2022), all the while maintaining user privacy. The detection of abnormal vehicle trajectories at traffic crossings has been accomplished through the use of FL in conjunction with OneClass Support Vector Machine (OC-SVM) Koetsier et al. (2022). According to the published results, the federated strategy enhances anomaly detection's overall accuracy while also benefiting specific data owners. In other where FL provided promising results is predicting steering wheel angles and traffic control. The performance of centralized learning and FL in steering angle prediction was evaluated in P et al. (2021) under various noise levels, and the outcomes were equivalent. This research also took into account the effects of communication load and interruptions, offering a thorough assessment of the systems. Because of this, FL is appropriate for applications that include a growing number of CAVs, particularly for jobs like steering wheel angle prediction. The research provided in Zhang et al. (2021) showed that using FL in CAV significantly improved the quality of the edge models. In particular, the research used optical flow and pictures as two data modalities to estimate steering wheel orientations. By employing FL to update the controller parameters dynamically, the target speed has been better achieved while improving driver comfort and safetyZeng et al. (2022). Moreover, FL is applied in cooperative parameter optimization between several vehicles at traffic junctions, preventing crashes and enhancing driving comfort Wu et al. (2021). By precisely calculating the road friction coefficients, FL is used in Liu et al. (2021) to improve brake performance in a variety of driving scenarios and settings. This method maximizes the braking action while protecting the driver's privacy. To optimize the controller design for AVs with variable vehicle participation in the FL training process, a FL framework is proposed in Zeng et al. (2022).

**Lable Flipping in Federated Learning.** The rising popularity of FL has led to the exploration of various attacks in this context, such as backdoor attacks Zhuang et al. (2024), gradient leakage attacks Yang et al. (2024), and membership inference attacks Zhu et al. (2024). In work we focus on data poisoning attacks[Data Poisoning Attacks Against Federated Learning Systems], such as label-flipping (LF) Li et al. (2023a), Biggio et al. (2012) and feature perturbation (FP), are critical areas of research. LF attacks have been widely applied in image processingPaudice et al. (2019). For instance, Nowroozi et al. (2023) evaluated LF attacks and proposed a defense mechanism using real datasets from the UCI repository , including MNIST Deng (2012) and Spambase Hopkins & Suermondt (1999). Further experiments Rosenfeld et al. (2020) manipulated the MNIST dataset using LF attacks and found a slight increase in classification error after injecting ten poisoning points. The study was repeated with Multiclass Logistic Regression, revealing an error increase from 2% to 2.1% due to a random LF attack. The approach in Tolpegin et al. (2020) extended to label-specific scenarios, where adversaries could adjust predictions based on predetermined rules. Experiments on CIFAR-10 and a reduced version of ImageNet confirmed the effectiveness of the proposed method. In order to protect against poisoning assaults, a number of studies concentrate on evaluating certain updates. To differentiate between faulty and accurate updates, Jebreel et al. (2020)[ suggest examining the biases in the output layer. But only in the IID setting does it take model poisoning attacks into account. In order to prevent data poisoning attacks, FGold Fung et al. (2020) and Awan et al. (2021) evaluate the weights of the output layer; But these techniques also frequently penalize identical but good updates mistakenly, which causes the model's performance to significantly decline. Using a kernel density estimator, Li et al. (2023b) calculates how harmful each local update is in relation to its k-nearest neighbors. After that, it uses an asymptotic threshold to determine if updates are benign or poisoned. Not only is it difficult to choose a threshold of this kind, but this approach has not been validated with big DL models or non-IID data. In order to identify the LF attack, Qayyum et al. (2022) suggests a method for discovering the correlation between the latent features of training data and updates. However, the strategy imposes an additional cost on all parties to train another model that learns such relationship. Furthermore, it is unrealistic to believe that throughout the early training rounds, all peers will behave appropriately.

# 3 STUDY DESIGN

For our experiment we used three SunFounder Picar Sunfounder that we trained using a CNN (Convolutional Neural Network) model. The collected data from the integrated camera where used to

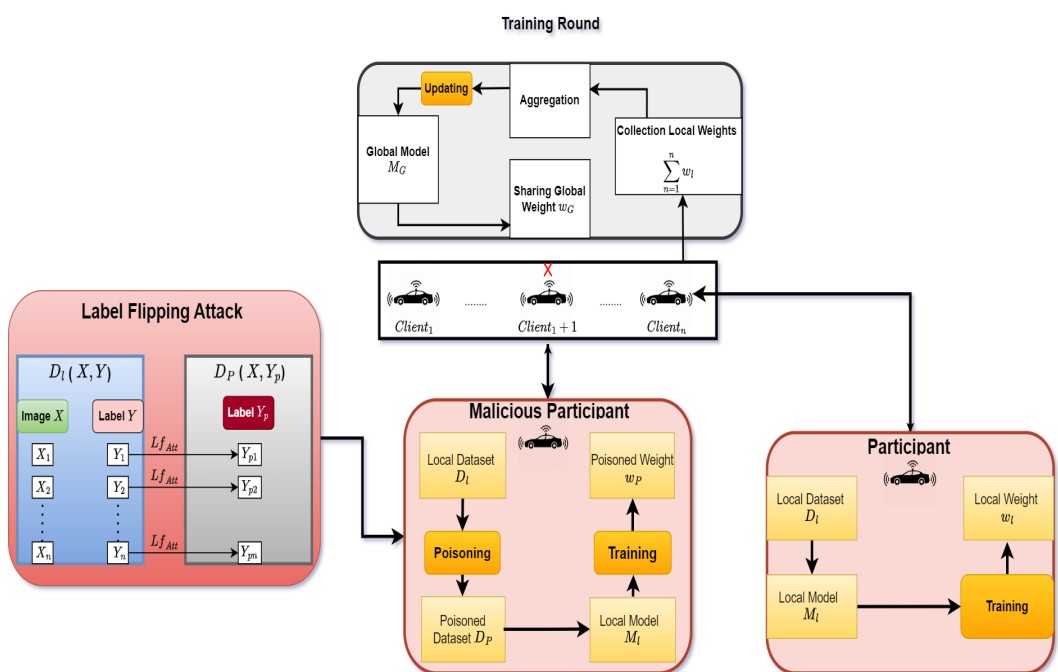

Figure 1: Proposed Federated Learning model architecture for Label flipping Attack on for AV

create our own dataset that contained 5K images of traffic light, lines and different types of obstacle. Moreover to address the data shortage we used the CIFAR10 dataset CIFAR-10 with it 60 K colored images of 10 different classes and we divided it into 50K data for training and 10K for testing and The A2D2 Dataset Audi that features over 40k labeled with 38 features.

We implement our FL framework for malicious vehicle detection using N = 3 participants, one central aggregator, and k = 5 each . We use an independent and identically distributed (iid ) data distribution, we assume the total training dataset is uniformly randomly distributed among all participants with each participant receiving a unique subset of the training data. The testing data is used for model evaluation only and is therefore not included in any participant Pi's train dataset. Observing that both CNN models converge after fewer than 200 training rounds, we set our FL experiments to run for R = 200 rounds total.

We trained our federated learning models on each dataset without adversarial settings. Next, the appropriate global models on test samples for each dataset. We first determined samples with high prediction confidence by computing softmax probabilities and choosing the examples that correctly forecasted in order to generate the Complementary dataset. If the projected class matched the actual label and its corresponding probability exceeded the threshold, we added the sample to the our dataset. Participants within the federated learning framework must maintain continuous communication and collaboration with the aggregation server. The model $M$ is finished with parameters $\theta_R$ at the end of $R$ rounds of FL. The test dataset used to evaluate $M$ is denoted by $D_{\text{test}}$, where $D_{\text{test}} \cap D_i = \emptyset$ for each participant dataset $D_i$. We present an in-depth analysis of label flipping attacks in FL in the following sections.

### 3.1 LABEL FLIPPING ATTACK

We use a label flipping attack to implement targeted data poisoning in FL. Given a source class $C_{\text{Source}}$ and a target class $C_{\text{target}}$ from C, each malicious participant Pi modifies their dataset Di as follows: For all instances in Di whose class is $C_{\text{Source}}$, change their class to $C_{\text{target}}$. We denote this attack by $C_{\text{Source}} \rightarrow C_{\text{target}}$. For example, images with initial red light class labels may be altered to have a green light class by malicious participants, according to the CIFAR-10 image classification sign red light $\rightarrow$ green light. The attack tries to increase the possibility that, during testing, the final global model would mistakenly classify traffic signals. The threat of label flipping is well-known in centralized machine learning. It's also acceptable in the FL scenario given the hostile objective

---

**Algorithm 1** Server-side code

---

**Require:** $n > 0$ the amount of clients
  $CNN \leftarrow create\_model()$
  **for** i in [0, n] **do**
    $CNN[i].initiate(initial\_packets)$
  **end for**
  **for** i in [0, n] in parallel **do**
    $open\_port()$
    $await\_client\_connection()$
    $CNN[i] \leftarrow receive\_client\_CNN()$
  **end for**
**Require:** $suspicious\_packet$
  **return** $CNN.predict(suspicious\_packet)$

---

**Algorithm 2** Client-side code

---

  $CNN \leftarrow create\_decision\_CNN()$
  $CNN.initiate(initial\_packets)$
  $CNN.train(local\_Data)$
  $connect\_to\_server()$
  $send\_CNN\_to\_server()$

---

and capabilities indicated above. Label flipping is different from other poisoning methods in that the adversary does not need to know the CNN architecture, loss function L, global distribution of D, etc. Its time and energy efficiency make it a desirable feature, especially since FL is frequently used with edge devices. In addition, it is simple enough for non-experts to perform and doesn't involve changing or meddling with participant-side FL software. To simulate the label flipping attack in a federated learning (FL) system with $P\ddot{} = 3$, where one is $P$ malicious, we proceed as follows. At the start of each experiment, we randomly designate $N \times m\%$ of the participants from the total $P$ as malicious, while the remaining participants are considered honest. The malicious participants is then injected with flipped labels, each experiment is repeated 10 times, and we report the average results. We examine three label flipping attack settings that represent a range of adversarial conditions:

- A source class $\rightarrow C_{\text{target}}$ class pairing where the source class was very frequently misclassified as the $C_{\text{target}}$ class in federated, non-poisoned training.

- A pairing where the source class was very infrequently misclassified as the $C_{\text{target}}$ class.

- A pairing between these two extremes.

These conditions provide a diverse set of scenarios to evaluate the effectiveness and impact of label flipping attacks in the federated learning environment.

### 3.2 ATTACK EVALUATION METRICS

We employ several evaluation indicators to do this.

**Global Model Accuracy** ($M_{\text{acc}}$): The global model accuracy is the percentage of instances $x \in D_{\text{test}}$ where the global model $M$ with final parameters $\theta_R$ predicts $M_{\theta_R}(x) = c_i$ and $c_i$ is indeed the true class label of $x$.

**Class Recall** ($c_{\text{recall}_i}$): Where the percentage

$$\frac{T_{P_i}}{T_{P_i} + F_{N_i}} \cdot 100\%$$

represents the class recall for any class $c_i \in \mathcal{C}$. Whereas $F_{N_i}$ is the number of examples $x \in D_{\text{test}}$ where $M_{\theta_R}(x) \neq c_i$ and the true class label of $x$ is $c_i$. The number of instances $x \in D_{\text{test}}$ is $T_{P_i}$, where $M_{\theta_R}(x) = c_i$ and $c_i$ is the true class label of $x$.

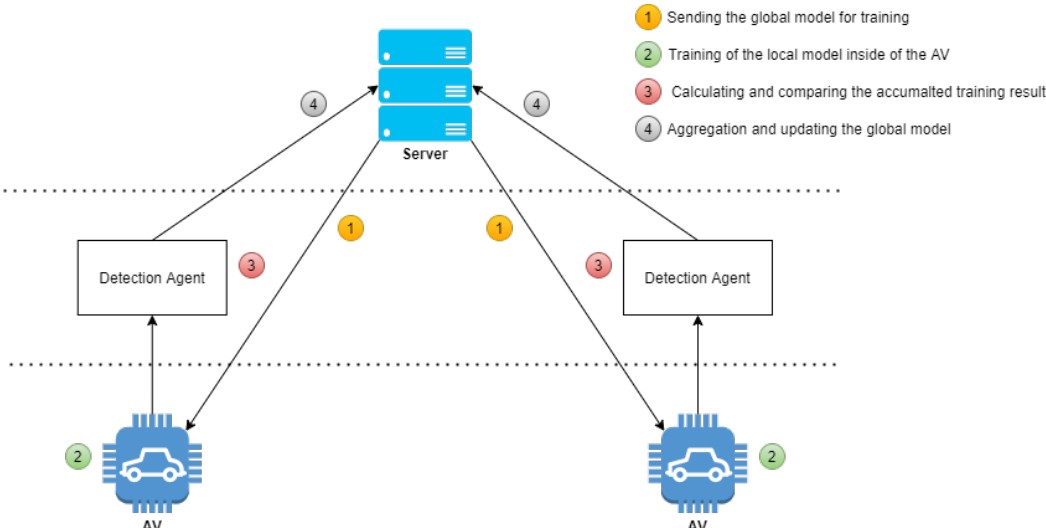

Figure 2: Overview of the system design

## 3.3 PROPOSED DEFENSE MECHANISM

The defense approach for the LF attack consist of creating a security layer in between the aggregation layer that includes our server $S$, where we trained our initial model and used for the distribution and updates the global model by aggregating locally learned models from the vehicles. the aim is to reach the optimal weight by optimizing the loss function L(W).

$$W_{\text{Op}} \leftarrow \arg\min \mathcal{L}(W),$$

The other layer is the training layer that consists of $3P$ vehicles, which, after obtaining a global model from the reliable server, jointly train the local model. Each $P$ trains a local model using its personal dataset $D_p \in D$.

Our defense mechanism consist of locating and extracting from each participant local updates the gradients pertinent to the LF attack, before updating the global model. This is made using the Principal Component Analysis (PCA) to reduce the dimensionality of update vectors and; Multiclass Classification Using Support Vector Machines (SVM). Our security agent study the relevant parameter subset and verify each peer update, since the updates provided by malicious participants differ from those sent by honest participants. After obtaining two dimension we use MLSVM to compute the distance between each sample point and the several class-specific hyperplanes, the model identifies the hyperplane and, thus, the class assignment. In an MCSVM with $M$ classes, the model calculates the decision function for each classifier for a given test sample by training $M$ binary classifiers, each of which corresponds to a distinct class. With the reduced dimension the objective is to differentiate the $M$ class with the all the different training data. Once each class is defined, we compare the local model with our initial global model by calculating the

$$\text{Outlier\_Score}_{t+1} = \frac{1}{|D_C|} \sum_{i=1}^{|D \sum C|} (y_i \neq \hat{y}_i)$$

the higher the score the more likely to be a malicious participant. Once the agent has determined which users are malicious, it may ban them or stop using their update.

---

**Algorithm 3** Federated Learning Algorithm with Adversarial Mitigation and PCA

---

**Require:** $P$: Total number of AV nodes
**Require:** $K$: Participating vehicles during aggregation
**Require:** $R$: Federation rounds
**Require:** $D_C$: Initial dataset
**Require:** $\mathcal{A}$: Adversary-controlled vehicles list
**Require:** $M_s$: MCSVM model
**Require:** $\mathbf{W}_l$: Local model
**Ensure:** $\mathbf{W}_t$: Global model shared with all peers in $R$- federation round
  1: $\mathbf{W}_0 \leftarrow$ initialize global model
  2: **for** $t = 0$ to $T - 1$ **do**
  3:     $S \leftarrow$ random set of $K$ participants (Client Identifier)
  4:     $C$ sends $\mathbf{W}_t$ to all participants in $S$
  5:     **for** each participant $k \in S$ in parallel **do**
  6:         **if** $k \in \mathcal{A}$ **then**
  7:             Poison data $D_k$ through LF
  8:         **end if**
  9:         $\mathbf{W}_{t+1}^k \leftarrow$ ClientUpdate$(k, \mathbf{W}_t)$
 10:     **end for**
 11:     **for** each participant $k \in S$ **do**
 12:         **Apply PCA:** Transform $\mathbf{W}_l$ using PCA to reduce dimensionality:

$$\mathbf{W}_l \ = \mathrm{PCA}_{\mathbf{W}_l}$$

 13:         **for** each sample $i$ in $W_l$ **do**
                 $M_s \ ( \ W_l \ )$
 14:         **end for**
 15:         Test the $\mathbf{w}$ model and compute $\hat{y}_{i,k}^L$ for each $M_s \ ( \ W_l \ )$
 16:         Test the model $\mathbf{W}_{t+1}^k$
 17:         Compute Outlier_Score$_{t+1}$
 18:     **end for**
 19:     $O \leftarrow$ Select $\tau$ participants with the highest Outlier_Score$_{t+1}$ scores
 20:     $G \leftarrow P - O$ {Remove malicious participants from $P$}
 21:     Perform aggregation of models $\mathbf{W}_{t+1}$
 22: **end for**
 23: **return** $\mathbf{W}_T$

---

### 3.4 EVALUATION INDICATORS

Throughout this paper, we employ metrics to enhance our understanding of both the security and utility provided by the models under scrutiny, which we subject to experimental manipulation. These metrics have been defined as follows:

- Source Class Recall: This metric calculates the number of correct positive predictions made out of all positive predictions that could have been made by the model. =In the event of label tampering by a malicious user, this metric will decrease, as fewer (or none) correct positive predictions will be made for the specific class $C_{\text{target}}$ by the attacker.

- Sparse Categorical Accuracy: This metric evaluates the accuracy of a model's predictions by comparing the predicted class labels with the actual ground truth labels.

- CrossEntropy Loss: This measures the disparity between the predicted probability distribution and the true probability distribution of the classes. In our models, it serves to quantify how well the predicted probabilities align with the actual class labels.

## 4 RESULTS

First, we study the effect of a single malicious participant on our federated learning framework, FL. We find that just a single malicious participant can significantly degrade global model performance-

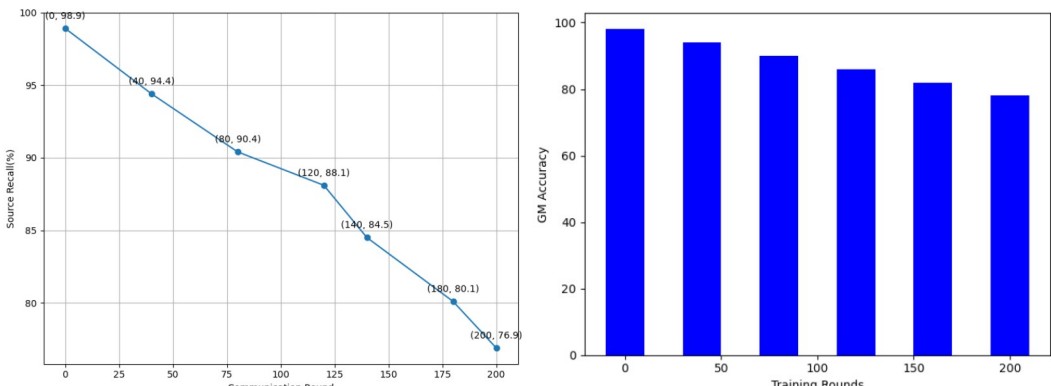

Figure 3: Evolution of the source class recall by round when $\alpha = 0.8$

Figure 4: Evaluation of the Global model accuracy with a malicious participant

source class recall losses of over 25% are possible when this adversary is consistently well-represented in the participant pool. This impact on source class recall is highest for a high availability level of $\alpha = 0.9$. In this respect, the effect becomes smaller if availability goes down, meaning here that lower losses can be obtained for lower values: $\alpha = 0.7$, $\alpha = 0.6$ or $\alpha = 0.5$. Thus, to maximize effectiveness of the attack it is beneficial for the malicious participant to remain as available as possible-especially in the later training rounds. To further demonstrate this effect of availability, we report source class recall by round for $\alpha = 0.7$ and $\alpha = 0.9$. A higher availability of the malicious participant results in a noticeable degradation in source class recall, along with lowering the value of recall for $\alpha = 0.9$ compared to $\alpha = 0.7$. The probabilistic selection of the participants can be considered one of the prime reasons for the variability of recall across rounds. A round with fewer malicious participants tends to increase source recall, and a higher number falls back.Each experimental condition is run three times, and the outcomes are averaged to remove round-to-round variability. As our results indicate, even a single malicious participant can significantly reduce global model performance, with source class recall losses of over 25% possible under high availability. Indeed, with high availability, there is a negative effect, while a decrease in availability tends to grant considerably better results. Importantly, for values of $k$ significantly larger than $N \times m\%$, increasing availability ($\alpha$) becomes less effective for meaningful impacts in individual training rounds. As for the accuracy we notice that each round of training using the malicious participant effected the model accuracy by a drop of 0.1% with each training round leading to an overall loss of 20% when to model finish training.

Using our proposed defense mechanism enables the detection such a malicious participant and never allows any updates from that participant or blacklists the participant for further usage in rounds. Leading to no accuracy lost and the integrity of the global model training. Using Principal Component Analysis (PCA) and Multi-class Support Vector Machine (SVM) classifiers together offers a strong way to defend against label flipping attacks in federated learning. PCA cuts down the number of features and gets rid of noise. This proves important in federated setups where data quality often changes from one user to another. Focusing on the most useful features helps the model work faster and spot unusual data more easily. After the data gets changed by PCA, the usage of Multiclass Classification SVM classifiers to draw complex lines is helpful for our agent in telling apart good labels from malicious ones. This method really works in finding strange data and keeping correct results even with malicious participation.

## 5 CONCLUSION

In this paper, we investigated data poisoning attacks targeting Federated Learning (FL) systems in autonomous vehicles. Our study reveals the susceptibility of FL systems to label flipping poisoning attacks, highlighting their significant adverse effects on the global model. We established a defence mechanism biased on PCA and MCSVM do to their ability to separate outliers. We further investigated the results of our mechanism which lead to the avoidance of the attack and the preservation of the global model integrity.

**Future Work:** Since this approach was based on a real life simulation with real cars the number of participant was limited to three do the high necessity of computation cost. we aim to extend our approach on multiple participant. Also the recreation of our strategy with a computer simulation using different nodes and compare it to stats ot the art solution. Another critical area is to test our defence strategy against other types of adversarial attack such as backdoor and noise injection. One other critical area to look into is domain adaptation since a participant can contain different data from the original but not harmless. Creating a distinguish between outlier and other domain data is crucial.

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
