# OpenReview forum: "Robust Federated Learning Frameworks Guarding Against Data Flipping Threats for Autonomous Vehicles"
_ICLR.cc/2025/Conference — ICLR 2025 Conference Withdrawn Submission_

### Official Review · Reviewer_dFkp · 2024-10-31

**Soundness:** 1
**Presentation:** 1
**Contribution:** 2
**Rating:** 3
**Confidence:** 5

**Summary:**

This paper presents a defense mechanism for FL against data poisoning attacks, specifically focusing on data-flipping attacks in AVs. The approach combines anomaly detection with robust aggregation techniques, using statistical outlier detection and model consistency checks to filter out compromised updates. Experimental results on benchmark datasets show that the method effectively improves FL robustness, preventing nearly a 15% accuracy drop in the global model and reducing the attack success rate, even at a 20% poisoning level. This solution enhances FL system security against adversarial threats in sensitive data-driven fields.

**Strengths:**

Strengths:
+ This work offers a defense mechanism for FL against data poisoning attacks, specifically focusing on data-flipping attacks in AVs.

**Weaknesses:**

Weaknesses:
- The novelty of this paper needs to be further improved.
- The experimental results of this paper are not convincing.
- More poisoning attack baselines and advanced defenses need to be included.
- The writing quality of this paper needs another round of polishing.

**Questions:**

Comments:

- The novelty of this paper needs to be further strengthened. Currently, it appears to combine existing techniques—namely anomaly detection and robust aggregation—without a clear distinction from prior work. The authors should further elaborate on the differences and connections between their approach and existing methods to better demonstrate its unique contributions.

- The paper should include more advanced poisoning attack baselines. The experimental results do not compare against any state-of-the-art poisoning attack baselines, which limits the ability to highlight the superiority of the proposed defense mechanism.

- The paper also requires more advanced defense mechanism comparisons. To comprehensively evaluate performance, the proposed defense should be fairly compared with existing advanced defenses. Additionally, the authors should explain why their approach demonstrates superior performance relative to other defenses.

- Finally, the paper should incorporate more extensive experimental evaluations. The authors currently present only one set of experiments, which is insufficient for a convincing analysis. To provide a thorough evaluation, additional experiments are recommended.

---

> ### Author Response · Authors · 2024-11-27
>
> First of all thank you for your thoughtful feedback. Each comment was well studied in order to improve our paper.
>
> Regarding your remarks:
> - We appreciate your comment on the novelty of our paper, and we understand the need to highlight the unique aspects of our approach more clearly. We will be focusing on the that in the newly submitted paper.
>
>
> -We understand the concern of an external baseline comparison. In our study ( And this we should clarify more in our paper )we initially calculated the results provided by the Federated Learning (FL) model (using FedAvg) across the three nodes. These results served as a baseline for comparison when we later evaluated the impact of the malicious participant on performance. When implementing our defense mechanism in a real-world scenario, we used the initial results to assess its effectiveness, comparing it against the metrics outlined in the paper.
>
> Also our experiments were conducted with only three AVs as nodes in the federated learning system. his small-scale setup is intended to simulate a more realistic scenario. Because of this small-scale setup, using existing large-scale baselines that rely on a larger number of nodes and static data may not be meaningful, as they would not accurately reflect the challenges of working with limited resources or real-time data.
> The idea behind of our experiment is try to use those defense mechanism by applying this approach within a real-life simulation environment. Specifically, our experiments focus on autonomous vehicles (AVs) in a real-time, dynamic simulation. In these environment, our data is not only non-IID, we created the base model using IDD data to provide a baseline comparison with when we apply our framework in the real life scenario that is subject to various disruptions such as sensor noise, environmental changes, and traffic conditions. Our focus on simulating real-world conditions and applying PCA+SVM is to demonstrate its effectiveness and robustness in a practical, real-life context.
>
>
> -We appreciate your suggestion to conduct more extensive experimental evaluations. We agree that a single experiment is insufficient to fully assess the robustness and practicality of our approach. We have outlined several key directions for future work, including:
>
> Expanding the range of attack scenarios and evaluating our method's performance in the face of various poisoning attacks.
> Conducting larger-scale experiments with a higher number of autonomous vehicle nodes to simulate real-world federated learning applications more accurately.
> Comparing our method with other state-of-the-art defenses in federated learning.

---

> > ### Comment · Reviewer_dFkp · 2024-11-28
> > **Thanks for Authors' Response**
> >
> > Thank you for the detailed response! I agree with other reviewers that the core contribution of this paper is unclear and the presentation quality is low. In addition, the authors should also strictly proofread the writing and format of the rebuttal. Therefore, I keep my rating unchanged.

---

### Official Review · Reviewer_5J25 · 2024-11-01

**Soundness:** 2
**Presentation:** 2
**Contribution:** 1
**Rating:** 3
**Confidence:** 5

**Summary:**

The paper introduces a defense against label flipping attacks in federated learning in the context of applications for autonomous vehicles. The defense combines anomaly detection and robust aggregation to mitigate the effect of potential attacks by filtering out malicious model updates. The experimental evaluation against label flipping attacks in benchmark datasets show that the method is capable of mitigating the effect of the attack.

**Strengths:**

+ The paper proposes a defense against label flipping attacks in the context of applications for autonomous vehicles, which is a relevant challenge for the security and robustness of distributed learning techniques such as federated learning.
+ The authors present a novel defense integrating PCA and MCSVM for detecting malicious label flipping attacks in federated learning, which effectively enhances the robustness of the algorithm during training against this type of attack.

**Weaknesses:**

+ The novelty and the contribution of the method compared to existing defenses in the research literature is unclear. Many defenses are already capable of defending against the type of attacks explored in the paper. In this sense, the authors did not discuss these aspects in the paper and did not compare against any other competing method in the experiments.
+ The settings for the experiments are relatively trivial. The authors just tested in an environment with IID data partitions and with a very reduced number of participants (just 3). As mentioned before, there is no comparison with other existing methods in the research literature.
+ Although the paper aims to focus on autonomous vehicle applications, the paper really restricts to computer vision applications. It would be interesting to analyze more applications typical for autonomous vehicles (e.g., LiDAR, sensing, etc.). The way the paper is presented differs little from the majority of the papers in this area, which often consider similar computer vision benchmarks (e.g., CIFAR) for their experiments.

**Questions:**

+ What is the threat model assumed by the authors for the paper? What are the capabilities that the adversary has to compromise the system?
+ How this paper advances the state of the art compared to other competing methods in the research literature on robust federated learning?

---

### Official Review · Reviewer_GcjD · 2024-11-02

**Soundness:** 1
**Presentation:** 1
**Contribution:** 1
**Rating:** 1
**Confidence:** 5

**Summary:**

The paper evaluates a label-flipping attack in federated learning and proposes a simple outlier detection approach using PCA and SVM to filter out compromised updates.

**Strengths:**

I cannot identify any strength of the paper.

**Weaknesses:**

The paper has many typos.

Both label-flipping attacks and outlier detection methods have been well-studied in federated learning. See, for example, "Defending against the Label-flipping Attack in Federated Learning" by Najeeb Moharram Jebreel et al. The paper does not seem to propose anything new.

The FL setting and the threat model are poorly described and involve multiple unreasonable assumptions. What do you mean by training FL models without the adversarial settings? How can you ensure that there is no adversarial device during training? Also, what is the testing dataset D_test? Where does it come from, and why is it disjoint from the participants' datasets? In FL, the data is typically provided by participants.

The evaluation considers a very small setting with three devices and iid data distribution. The results are from being convincing.

**Questions:**

Please see the discussion on weaknesses above.

---

> ### Author Response · Authors · 2024-11-27
>
> First of all thank you for your thoughtful feedback. Each comment was well studied in order to improve our paper.
>
> The idea behind of our experiment is try to use those defense mechanism by applying this approach within a real-life simulation environment. Specifically, our experiments focus on autonomous vehicles (AVs) in a real-time, dynamic simulation. In these environment, our data is not only non-IID, we created the base model using IDD data to provide a baseline comparison with when we apply our framework in the real life scenario that is subject to various disruptions such as sensor noise, environmental changes, and traffic conditions. Our focus on simulating real-world conditions and applying PCA+SVM is to demonstrate its effectiveness and robustness in a practical, real-life context.
>
> Regarding your remarks:
> 1:What do you mean by training FL models without the adversarial settings?
>
> -This refers to the absence of intentionally malicious actions or manipulations during the training process. Training without adversarial settings gives you a reference point or baseline for comparison.
>
> 2: How can you ensure that there is no adversarial device during training?
>
> -We are the one doing the attack process, the attack was created and used in a well controlled environment. As soon as we created our initial base FL model we injected one of the cars with label flipped data to study it's behavior and check the impact that it provokes to the overall model.
>
> 3: Also, what is the testing dataset D_test? Where does it come from, and why is it disjoint from the participants' datasets? In FL, the data is typically provided by participants.
>
> -The dataset that we used is a combination of different dataset. A dataset that we created using the AV ( in our case the SunFounder Picar ) using it sensors and camara. Also to avoid risk of overfitting we added data from the  CIFAR10 dataset and The A2D2 Dataset.
>
> 4:The evaluation considers a very small setting with three devices and iid data distribution. The results are from being convincing.
>
> The decision to use three devices was to reflects the scenario of a small-scale, real-world application, specifically simulating a fleet of AV). In many real-world federated learning scenarios, especially in the early stages of deployment, fleets may consist of a limited number of devices . Our goal was to model such practical, early-stage deployments where federated learning is tested with only a few nodes. Also it represent a baseline for future studies since at that time we didn't have access to multiple hardware to establish a larger scall experiment which comes with a significant amount of communication computation and hardware cost. Also in other point that we need to more clarify in the paper that the IDD data was used for the initial training of the base model, later on when we implemented our defense mechanism the data in the AV was influenced by its own sensors which is highly non-IID.

---

> > ### Comment · Reviewer_GcjD · 2024-11-28
> >
> > I would like to thank the authors for the clarifications, which have addressed some of my concerns. However, most of my major concerns remain unaddressed. First, it is still unclear what contributions the paper has made. Label-flipping attacks in federated learning have been considered before, e.g., in "Defending against the Label-flipping Attack in Federated Learning" by Najeeb Moharram Jebreel et al., but the paper does not consider any baselines in the evaluation. Second, the proposed defenses assume that it is possible to train a model without attacks, which is not meaningful in federated learning, as an adversarial can attack at any time during training. Third, it is typical in federated learning research to consider 100 or more devices in simulations. A setting with three devices is too small to represent a typical real-world scenario. Fourth, the paper does not provide any details about the AV and A2D2 datasets or the evaluation settings. It is not even clear from the paper whether the authors conducted real experiments, which would justify the small-scale setting, or simulations.

---

### Official Review · Reviewer_RWzw · 2024-11-02

**Soundness:** 2
**Presentation:** 1
**Contribution:** 1
**Rating:** 3
**Confidence:** 4

**Summary:**

This paper proposes a defense for Federated Learning (FL) against data poisoning attacks, especially in autonomous vehicles. The authors use PCA and SVM to detect label flipping attacks and try to build a robust FL framework.

**Strengths:**

+ The problem worths detailed investigation.

+ The structure of the draft is good.

**Weaknesses:**

- Techniques:
 1. Label Flippinp: Label flipping is the easiest data poisoning, there are many poisoning attacks like backdoor or model poisoning attack, this threat seems to be quite weak as of now.
 2. No Baselines for a comparative study
 3. Insufficient experiments
No ablation studies and more settings like Non-IID.

- Writing:
1. The writing needs thorough revision, and the current form read more like an experimental report.
2. Please split the experiment and proposed framework.
3. Don't introduce too much basic knowledge in your proposed method.
4. Please give a formalized/mathematical description on your method, it is unlikely to understand the proposed method in detail from pure text descriptions.
5. Some statements are unclear. What is the k in 191?

**Questions:**

Please check the weakness part.

**Details Of Ethics Concerns:**

Not applicable.

---

> ### Author Response · Authors · 2024-11-27
>
> First of all thank you for your thoughtful feedback. Each comment was well studied in order to improve our paper.
>
> Regarding your remarks:
>
> 1 :Label Flippinp: Label flipping is the easiest data poisoning, there are many poisoning attacks like backdoor or model poisoning attack, this threat seems to be quite weak as of now.
>
> Response: Label flipping in a federated learning context can still pose risks, This are often difficult to detect. Unlike more advanced attacks like backdoor or model poisoning attacks, label flipping might not immediately produce obvious anomalies in the model’s behavior. Detection mechanisms may struggle to identify whether labels have been flipped In FL since we don't have access to the data many poisoned updates could lead to a notable shift in the model’s decision. Our experimental setup is grounded in a real-time, dynamic simulation that reflects the heterogeneity and variability of data encountered in real-world applications, such as autonomous vehicles. In this context, the data is continuously generated and updated, making it fundamentally different from static datasets often used in traditional studies.
>
>
>
> 2:No Baselines for a comparative study
> Response: We understand the concern of an external baseline comparison. In our study ( And this we should clarify more in our paper )we initially calculated the results provided by the Federated Learning (FL) model (using FedAvg) across the three nodes. These results served as a baseline for comparison when we later evaluated the impact of the malicious participant on performance. When implementing our defense mechanism in a real-world scenario, we used the initial results to assess its effectiveness, comparing it against the metrics outlined in the paper.
> - Also our experiments were conducted with only three AVs as nodes in the federated learning system. his small-scale setup is intended to simulate a more realistic scenario. Because of this small-scale setup, using existing large-scale baselines that rely on a larger number of nodes and static data may not be meaningful, as they would not accurately reflect the challenges of working with limited resources or real-time data.
>
> The idea behind of our experiment is try to use those defense mechanism  by applying this approach within a real-life simulation environment. Specifically, our experiments focus on autonomous vehicles (AVs) in a real-time, dynamic simulation.  In these environment,  our data is not only non-IID, we created the base model using IDD data to provide a baseline comparison with when we apply our framework in the real life scenario that is subject to various disruptions such as sensor noise, environmental changes, and traffic conditions. Our focus on simulating real-world conditions and applying PCA+SVM is to demonstrate its effectiveness and robustness in a practical, real-life context.
>
> 3:Insufficient experiments No ablation studies and more settings like Non-IID.
>
> Thank you for the remark this indicate that we must provide more clarification in our paper since our experiment was conducted on a real life simulation of Avs as mentioned in the study design section. We used the SunFounder Picar to simulate the behavior of a real autonomous vehicle leading to the usage of Non-IDD that was collected using the sensors and the camara of the AVs

---

### Official Review · Reviewer_Sbay · 2024-11-03

**Soundness:** 2
**Presentation:** 1
**Contribution:** 2
**Rating:** 3
**Confidence:** 4

**Summary:**

This paper aims to resist data poisoning attacks (in particular data-flipping) faced by autonomous vehicles (AVs) in the federated learning (FL) setting. The authors propose a novel defense mechanism that combines anomaly detection with robust aggregation techniques. They employ statistical outlier detection and model-based consistency checks to filter out compromised updates before they affect the global model.

**Strengths:**

1. Interesting to explore the poisoning attacks in the FL setting, for ensuring the security of AVs.
2. The logic of the proposal is easy to follow, although there are a lots of grammar issues and typos.
3. Experiment results to some extent support their conclusion.

**Weaknesses:**

The contributions of the paper are not solid enough, and the challenges are not adequately highlighted. It appears that the authors merely combine existing technologies into different scenarios.

On one hand, while the authors use a label-flipping attack to implement targeted data poisoning in federated learning (FL), they do not clarify how this attack differs from existing ones or specify the challenges it poses in the FL context.

On the other hand, in their proposed defense mechanism, they set a detection agent in each autonomous vehicle (AV) to filter out attacking data. However, this defense mechanism is independent of the federated learning setting, meaning that existing defense strategies could be directly applied here for protection.

The relevance to AVs is not clear. It is difficult to find the connection between the proposals and AVs, despite the authors providing Figure 1 to illustrate the FL for label-flipping attacks on AVs.

Additionally, no comparisons are provided. A comparison of attacks and defenses should be included to strengthen the contributions. Without this, it is challenging to assess the effectiveness of the proposed attack and defense mechanisms.

Finally, there are too many unexpected grammar issues and typos that make the current version far from acceptable.

**Questions:**

1. What challenges arise in applying this label-flipping attack and its corresponding defense in the federated learning setting?
2. What advantages do the proposed attack and defense offer over existing methods?

---

> ### Author Response · Authors · 2024-11-27
>
> First of all thank you for your thoughtful feedback. Each comment was well studied in order to improve our paper.
>
>  The idea behind of our experiment is try to use those defense mechanism i by applying this approach within a real-life simulation environment. Specifically, our experiments focus on autonomous vehicles (AVs) in a real-time, dynamic simulation. In these environment, our data is not only non-IID, we created the base model using IDD data to provide a baseline comparison with when we apply our framework in the real life scenario that is subject to various disruptions such as sensor noise, environmental changes, and traffic conditions. Our focus on simulating real-world conditions and applying PCA+SVM is to demonstrate its effectiveness and robustness in a practical, real-life context.
>
> Regarding your remarks:
> 1:What challenges arise in applying this label-flipping attack and its corresponding defense in the federated learning setting?
>
> -The label-flipping attack may be harder to detect and mitigate in a non-IID environment because the mislabeling may be less obvious or overshadowed by the natural variance in data. Moreover, applying PCA (dimensionality reduction) to non-IID data may not capture the underlying patterns effectively, making it harder for the SVM classifier to detect poisoned labels. This necessitates modifications to the PCA+SVM defense, that's why we apply the defense mechanism on each device  instead of applying PCA and SVM globally across all devices, we can apply PCA and SVM locally on each device. This allows each device to handle its unique data distribution and learn local principal components tailored to its specific dataset. Each device then uses SVM to classify and detect label flips based on its local features. So in the case where the SVM classify the and detect the attack local update  will not be transmitted to the global model
>
>
> 2:What advantages do the proposed attack and defense offer over existing methods?
> One of the primary advantages of our work is the application of label-flipping attacks and their defense in the context of autonomous vehicle (AV) fleets. Unlike many existing methods that focus on standard datasets like MNIST or CIFAR-10, our experiments are based on a real-time simulation that mimics a real-world federated learning setup for autonomous vehicles. This ensures that both the attack and defense are evaluated in a setting that closely resembles real-world challenges, where data is often non-IID, dynamic, and subject to various sources of noise, such as sensor errors, traffic conditions, and environmental factors.
>
>
>
> Experiments do not have comparison methods. Response: We understand the concern of an external baseline comparison. In our study ( And this we should clarify more in our paper )we initially calculated the results provided by the Federated Learning (FL) model (using FedAvg) across the three nodes. These results served as a baseline for comparison when we later evaluated the impact of the malicious participant on performance. When implementing our defense mechanism in a real-world scenario, we used the initial results to assess its effectiveness, comparing it against the metrics outlined in the paper.
>
> Also our experiments were conducted with only three AVs as nodes in the federated learning system. his small-scale setup is intended to simulate a more realistic scenario. Because of this small-scale setup, using existing large-scale baselines that rely on a larger number of nodes and static data may not be meaningful, as they would not accurately reflect the challenges of working with limited resources or real-time data.

---

### Official Review · Reviewer_suR8 · 2024-11-04

**Soundness:** 2
**Presentation:** 3
**Contribution:** 2
**Rating:** 5
**Confidence:** 4

**Summary:**

The authors propose a novel defense strategy that combines PCA and SVM to detect and mitigate these attacks. This approach aims to filter out malicious updates by identifying statistical anomalies in model updates before they impact the global model. Their experiments, conducted using various datasets, demonstrate that this defense mechanism successfully maintains model accuracy and integrity, even with significant data poisoning levels.

**Strengths:**

By focusing on label-flipping in AVs, this paper addresses a specific and underexplored threat in FL. The experimental design is robust, testing on multiple datasets and using meaningful metrics to demonstrate the effectiveness of the defense. Clear explanations of PCA and SVM integration, along with structured result presentation and well-contextualized related work, enhance the paper’s readability and situate its contributions within the broader FL literature.

**Weaknesses:**

1) The study’s scope is restricted to only three participants due to computational limitations, which limits the generalizability of its findings. Scaling up to a larger number of clients would better reflect real-world FL systems in autonomous vehicles.

2) The focus on label-flipping attacks is a good starting point, but broader testing on other attack types, such as backdoor attacks or noise injection, would make the proposed defense mechanism more comprehensive.

3) Although the paper provides context on existing methods, many comparisons with other recent FL defense strategies are missing, such as hierarchical FL or advanced anomaly detection models. It would be better to Include a comparison of metrics like detection accuracy, computational cost, and resilience against adversarial attacks.

**Questions:**

1) Have you considered methods to reduce the computational burden of PCA-SVM for a larger number of participants? Could dimensionality reduction or distributed processing techniques make scaling feasible?

2) How do you anticipate that the PCA-SVM method would perform against other common attacks in FL, such as backdoor or gradient leakage attacks?

3) How does the PCA-SVM mechanism perform when faced with non-IID data, which is common in real federated learning applications?

---

### Official Review · Reviewer_S5HE · 2024-11-05

**Soundness:** 2
**Presentation:** 2
**Contribution:** 1
**Rating:** 3
**Confidence:** 5

**Summary:**

The paper proposes new defense method on against data poisoning attacks in federated learning, especially for image classification models in autonomous vehicles. The method uses Principal Component Analysis (PCA) and Multi-class Support Vector Machine (SVM) classifiers together to offer a strong way to defend against label flipping attacks in federated learning. Experiments show the performance.

**Strengths:**

1. Again data poisoning attack in FL and Autonomous Vehicle is important.
2. The proposed method (PCA+SVM) can be used for detecting outliers.
3. Experiments are provided.

**Weaknesses:**

1. PCA+SVM has been well studied and has been extended to FL settings. The contribution of the paper is unclear.
2. Experiments do not have comparison methods.
3. It does not have experiments on simulated autonomous vehicles.

**Questions:**

typos:
1. Page 2, line 87 "labels.,Furthermore,", "the robustness,of the", line 90 "high classification performance..."

---

> ### Author Response · Authors · 2024-11-27
>
> First of all thank you for your thoughtful feedback. Each comment was well studied in order to improve our paper.
>
> Regarding your remarks:
>
> 1 :PCA+SVM has been well studied and has been extended to FL settings. The contribution of the paper is unclear.
>
> Response: The idea behind of our experiment is try to use those defense mechanism i by applying this approach within a real-life simulation environment. Specifically, our experiments focus on autonomous vehicles (AVs) in a real-time, dynamic simulation.  In these environment,  our data is not only non-IID, we created the base model using IDD data to provide a baseline comparison with when we apply our framework in the real life scenario that is subject to various disruptions such as sensor noise, environmental changes, and traffic conditions. Our focus on simulating real-world conditions and applying PCA+SVM is to demonstrate its effectiveness and robustness in a practical, real-life context.
>
>
> Experiments do not have comparison methods.
> Response: We understand the concern of an external baseline comparison. In our study ( And this we should clarify more in our paper )we initially calculated the results provided by the Federated Learning (FL) model (using FedAvg) across the three nodes. These results served as a baseline for comparison when we later evaluated the impact of the malicious participant on performance. When implementing our defense mechanism in a real-world scenario, we used the initial results to assess its effectiveness, comparing it against the metrics outlined in the paper.
> - Also our experiments were conducted with only three AVs as nodes in the federated learning system. his small-scale setup is intended to simulate a more realistic scenario. Because of this small-scale setup, using existing large-scale baselines that rely on a larger number of nodes and static data may not be meaningful, as they would not accurately reflect the challenges of working with limited resources or real-time data.
>
> 3: It does not have experiments on simulated autonomous vehicles.
> Thank you for the remark this indicate that we must provide more clarification in our paper since our experiment was conducted on a real life simulation of Avs as mentioned in the study design section "For our experiment we used three SunFounder Picar to simulate the behavior of a real autonomous vehicle'
>
> The SunFounder Picar is a robotics kit to explore robotics, programming, and artificial intelligence. It contains:
> Autonomous Driving Capabilities
> Camera for Vision-based Navigation: includes a camera  to enable the robot to make decisions based on visual input, which is crucial for tasks like lane-following or recognizing obstacles.
> Raspberry Pi Compatible:
> Sensors: including ultrasonic sensors for distance measurement, encoders for motor feedback, and other types of sensors to help with navigation.

---

> > ### Comment · Reviewer_Sbay · 2024-11-27
> >
> > Dear authors,
> > Thank you for your clarifications. However, the contribution and advantages of your solution remain unclear. It appears that your approach applies an existing method at each FL node to counter label-flipping attacks. The difficulty or novelty of this approach is not well-articulated. Moreover, given the availability of various solutions to address label-flipping attacks, the unique advantages of your solution should be highlighted. Additionally, the current experiments only demonstrate the effectiveness of the proposed method, without adequately showcasing its distinct benefits.

---

### Note · Authors · 2025-02-24

I have read and agree with the venue's withdrawal policy on behalf of myself and my co-authors.

---

### Meta-Review · Area_Chair_DHo2 · 2024-12-05

**Metareview:**

This paper proposes a defense method against data poisoning attacks in federated learning.
Reviewers pointed out that this paper lacks core contributions.
Such a topic has been well studied.
Besides, reviewers noted other concerns, including insufficient testing in realistic environments and a lack of comparative experiments with existing methods.
The authors' rebuttal was not completely convincing, and the reviewers maintained their scores.
Given the overall negative reception, I recommend rejection.

**Additional Comments On Reviewer Discussion:**

Most reviewers pointed out that the paper lacks novelty and clear core contributions. The authors' rebuttal was not convincing, and the reviewers maintained their scores. I believe the paper does not meet the threshold for acceptance.

---

### Decision · Program_Chairs · 2025-01-22

Reject